# Selective Loading and Variations in the miRNA Profile of Extracellular Vesicles from Endothelial-like Cells Cultivated under Normoxia and Hypoxia

**DOI:** 10.3390/ijms231710066

**Published:** 2022-09-02

**Authors:** Anny Waloski Robert, Bruna Hilzendeger Marcon, Addeli Bez Batti Angulski, Sharon de Toledo Martins, Amanda Leitolis, Marco Augusto Stimamiglio, Alexandra Cristina Senegaglia, Alejandro Correa, Lysangela Ronalte Alves

**Affiliations:** 1Stem Cells Basic Biology Laboratory, Instituto Carlos Chagas—ICC-FIOCRUZ/PR, Rua Professor Algacyr Munhoz Mader, 3775, Curitiba 81350-010, PR, Brazil; 2Gene Expression Regulation Laboratory, Instituto Carlos Chagas—ICC-FIOCRUZ/PR, Rua Professor Algacyr Munhoz Mader, 3775, Curitiba 81350-010, PR, Brazil; 3Core for Cell Technology-School of Medicine, Universidade Católica Paraná-PUCPR, Curitiba 80215-901, PR, Brazil; 4National Institute of Science and Technology for Regenerative Medicine (INCT-REGENERA), Rio de Janeiro 21941-902, RJ, Brazil

**Keywords:** extracellular vesicles, hypoxia, miRNA, CD133^+^ cells, endothelial-like cells

## Abstract

Endothelial-like cells may be obtained from CD133^+^ mononuclear cells isolated from human umbilical cord blood (hUCB) and expanded using endothelial-inducing medium (E-CD133 cells). Their use in regenerative medicine has been explored by the potential not only to form vessels but also by the secretion of bioactive elements. Extracellular vesicles (EVs) are prominent messengers of this paracrine activity, transporting bioactive molecules that may guide cellular response under different conditions. Using RNA-Seq, we characterized the miRNA content of EVs derived from E-CD133 cells cultivated under normoxia (N-EVs) and hypoxia (H-EVs) and observed that changing the O_2_ status led to variations in the selective loading of miRNAs in the EVs. In silico analysis showed that among the targets of differentially loaded miRNAs, there are transcripts involved in pathways related to cell growth and survival, such as FoxO and HIF-1 pathways. The data obtained reinforce the pro-regenerative potential of EVs obtained from E-CD133 cells and shows that fine tuning of their properties may be regulated by culture conditions.

## 1. Introduction

Extracellular vesicle (EV) is the broad term for particles delimited by a lipid bilayer that is naturally released from the cell and unable to replicate, i.e., does not contain a functional nucleus [1]. EVs encompass a heterogeneous population of particles including microvesicles, which are assembled and released from the plasma membrane, and exosomes that are of endosomal origin and released by exocytosis of multivesicular bodies [2,3]. EVs are released essentially by all types of cells and can be isolated from the supernatant of cell and tissue cultures [4,5], as well as most body fluids, such as breast milk [6] and urine samples [7]. Initially thought of exclusively as cell disposal bags, EVs are now known as a regulated means of communication over short and long distances between cells and tissues [3,8].

The EVs cargo is composed of bioactive factors, such as lipids, proteins, and all types of nucleic acids (including mRNAs and non-coding RNAs) and is modulated by specific stimuli and environmental conditions [4,9], or can be manipulated by the artificial addition of elements of interest [10]. The EVs composition also reflects their cellular origin, hence the importance of studying vesicles from different types of cells and under different physiological and pathological conditions [11,12]. The EV components can participate not only in various physiological processes such as homeostasis, inflammation, immunological interactions, and angiogenesis, but also in pathophysiological conditions involved in the development of cancer and metastasis, autoimmune diseases, and infections of various types [13,14,15,16,17,18]. EVs isolated from stem and endothelial progenitors have been tested in several disease models and shown to protect and/or repair kidney, nervous and heart tissue after injury, among other tissues and organs [19,20,21,22,23,24,25,26]. On the downside, however, EVs can also help in preparing the niche for metastases, for example [27,28].

Considered a promising EV source for therapeutic applications, endothelial progenitor cells (EPCs) and endothelial-like cells may be obtained by expanding in vitro CD133^+^ cells isolated from human umbilical cord blood (hUCB). EPC and hematopoietic stem cells have a common embryonic precursor sharing various markers, such as CD133 and CD34 [29]. The early EPC phenotype represented by the expression of CD133, among others, showed better proliferative activity and greater distribution of primitive progenitors than other studied populations [30]. Early EPCs, also known as myeloid angiogenic cells (MACs), promote angiogenesis in a paracrine fashion, whereas late EPCs, also called endothelial colony-forming cells (ECFC), participate more directly in the formation of new vasculature [31,32]. An enriched population of early progenitors can be obtained prior to cultivation by magnetic cell sorting based on positive selection of CD133^+^ [33]. Then, during successive passages in an inductive culture medium, expanded CD133^+^ cells lose expression of CD133 and start to acquire a phenotype closer to adult endothelial cells expressing markers such as CD31, vWF and CD146 [23,29].

Our group had previously shown that expanded CD133^+^ cells (E-CD133) significantly improved cardiac function in rats after severe myocardial infarction [21]. The improvement observed was mostly due to the activity of their secretome, which seems to be related to the induction of angiogenesis [21]. An in-depth proteomic analysis was conducted to identify the protein content in EVs isolated from E-CD133 compared to vesicles obtained from bone marrow-mesenchymal stromal stem cells (BMSCs). The protein profiles revealed that E-CD133 EVs are somewhat similar to BMSCs EVs, but also different enough to reflect their main paracrine effect, that is, inductors and/or modulators of angiogenesis, widely reported in vitro and preclinical studies [11].

The conditions to which the cells are submitted may also influence the EVs production [34], EVs yield [9,35], and affect their content [36,37,38], including the miRNA cargo [9]. To improve our knowledge of E-CD133 and the potential of their EVs, here we explored the miRNA content of E-CD133 EVs isolated from cells kept in both normoxia (N-EVs) and hypoxia (H-EVs). Our objective was to determine which miRNAs may be enriched in EVs compared to the miRNA retained in cells, indicating a specific regulation of miRNA loading in EVs.

## 2. Results and Discussion

### 2.1. Protein and RNA Yield from EVs Isolated under Normoxic and Hypoxic Conditions

The immunophenotype of the E-CD133 cells was previously determined by flow cytometry by Angulski and coworkers (2017) at the same passage as when the EVs were isolated: E-CD133 cells were positive for CD105, CD146, CD309, CD31 and vWF; and had low expression of CD45, CD14, CD34 and CD133 [11]. The immunophenotype of these cells is compatible with what was previously described for CD133^+^ cells after expansion [23]. Additionally, the immunophenotype observed is compatible with that expected for ECFCs, which are characterized by being CD31^+^, CD105^+^, CD146^+^, CD45^−^ e CD14^−^, and by their angiogenic potential, both intrinsic (potential to form new vessels) and through secretion of pro-angiogenic elements (paracrine effect) [31].

Initially, we investigated whether the different culture conditions (normoxia and hypoxia) would interfere with E-CD133 cell viability. Thus, after the EV collection procedure, the cells were stained with Annexin V and 7-AAD. The percentages of E-CD133 cells in apoptosis/cell death cultivated under normoxia and hypoxia were 15.7% and 13.81%, respectively (Appendix A).

The EVs showed typical characteristics of EV size and morphology independently of culture conditions, as confirmed by transmission electron microscopy and NTA analysis. The mode of E-CD133 N-EVs and H-EVs was 172.2 ± 13.68 nm and 168.7 ± 1.92 nm, respectively (Figure 1a). The quantification of protein and RNA content in EVs showed that the EV protein/cell ratio was lower for the hypoxia condition, but there was no statistical difference in the RNA yield/cell or RNA content/protein content from N-EVs and H-EVs isolated from E-CD133 (Figure 1b–d).

### 2.2. Hypoxia Treatment Changes the miRNA Content of E-CD133 Cells and EVs

Next, we performed RNA-Seq analysis to investigate the miRNA content of E-CD133 cells and EVs. The analysis of the identified miRNAs showed that there are some differences in the miRNAs content among samples from different donors (Appendix A). Principal component analysis grouped samples by type (cells or EVs) (Figure 2a) and by condition (hypoxia or normoxia) in EV samples only (Appendix A). Although the cell samples in the two conditions showed few differences, these are not enough to clearly discriminate them using PCA (Appendix A). We considered only miRNAs identified in at least two of the three donor samples in each condition (E-CD133 N–cells, H-cells, N-EVs and H-EVs) (Appendix A). An overall evaluation showed that we identified more miRNAs in hypoxia (136 and 115 for cells and EVs, respectively) in comparison with normoxia (102 for cells and 91 for EVs). Most of the miRNAs (>60%) were identified in normoxia and hypoxia, for cells and EVs, but some were exclusively found in one or other conditions (Figure 2b). Interestingly, the top five miRNAs with higher counts per million (CPM) were similar among cells and EVs from hypoxia and normoxia (Appendix A). miR-99b-5p, miR-10b-5p and miR-100-5p are among these miRNAs with higher CPM, and we confirmed their presence by RT-qPCR (Appendix A).

We considered as differentially enriched (DE) the miRNAs exclusively identified in one condition (qualitative analysis) and those differentially expressed (log2(FC) < −1 or >1, FDR < 10%). The comparison of the miRNAs cell content showed that 55 miRNAs were DE between normoxia and hypoxia, from which 11 and 44 were exclusively found in N-cells and H-cells, respectively (Figure 2b, Appendix A). Regarding the composition of EVs (N-EVs vs. H-EVs), we observed that most miRNAs (84) were found in both conditions, but 39 were DE (Figure 2b, Appendix A). Seven miRNAs were found only in N-EVs, 31 were exclusively found in H-EVs and one miRNA was differentially expressed: the miR-486-5p, which was downregulated in H-EVs (log2(FC) = −1.95, FDR = 10%) (Appendix A).

The miR-486-5p was already identified in exosomes from other endothelial progenitor cells, such as hUCB-derived ECFCs and human placental microvascular endothelial cells, both in normoxia or in a model of hypoxia/reoxygenation [39,40]. Although this miRNA was predominantly studied in an oncological context [41], many studies have demonstrated the potential effects of this miRNA in EVs. mir-486-5p delivered by EVs in a model of mice with ischemic kidney injury was able to attenuate the injury, decreasing PTEN expression and increasing phosphorylated Akt [39]. Recently it was demonstrated that hypoxia increased the abundance of miR-486-5p in small EVs from monkey MSC and that normoxia-derived small EVs overloaded with miR-486-5p had similar benefits, promoting angiogenesis and cardiac recovery in a nonhuman primate myocardial infarction model [42]. In our study, the E-CD133 EVs showed significant expression of miR-486-5p in normoxia (in comparison with H-EVs), which was confirmed by qPCR analysis (Appendix A).

### 2.3. Favoring Loading in EVs or Cell Retention Concur to Differential N- and H-EVs Cargo, Indicating Selective miRNA Sorting to EVs

Next, we compared the miRNAs enriched in the EVs or retained in the cells during hypoxia and normoxia (N-EVs vs. N-cells; H-EVs vs. H-cells). We considered enriched in the EVs the miRNAs exclusively found in EVs or with a log2(FC) > 1 (vs. cells). The miRNA exclusively identified in the cells or with a log2(FC) < −1 were considered to be retained in the cells. After hypoxia treatment, 30 miRNAs were enriched in H-EVs, among which 21 were exclusives and 11 had a log2(FC) > 1 (Figure 2c,d, Appendix A); and 51 miRNAs were retained within the H-cells (41 exclusive and 10 with log2(FC) < −1) (Figure 2c–d, Appendix A). During normoxia, 28 miRNAs were enriched in N-EVs, including 22 were exclusives and seven had log2(FC) > 1 (Figure 2c,e, Appendix A). On the other hand, 40 were retained within the cells, among which 33 were exclusives and 8 had log2(FC) < −1 (Figure 2c,e, Appendix A).

Notably, 14 miRNAs were found to be enriched in both EVs, and 16 were retained in the cells both in normoxia and hypoxia culture conditions (Figure 3a, Appendix A). Moreover, 14 miRNAs were enriched in the EVs only in hypoxia, whereas 12 were enriched in the EVs only in normoxia. For cells, 22 and 33 miRNAs were retained only in normoxia and only in hypoxia, respectively (Figure 3a). By RT-qPCR analysis, we confirmed the presence of miRNAs enriched in N-EVs (miR-127-3p, miR-484), H-EVs (let-7f-5p, miR-181b-5p) or in both N and H-EVs (miR-12136, miR-3135b, miR-423-5p, miR-486-5p). Notably, miR-12136, miR-3135b and miR-423-5p had the highest fold to cel-miR-39 in our analysis, suggesting that they are indeed highly expressed in both H- and N-EVs (Appendix A).

As described above, 39 miRNAs were DE when we compared EVs from hypoxia and normoxia (H-EVs vs. N-EVs), among which 31 were enriched in H-EVs and eight in N-EVs (Figure 2b). Then, we investigated the miRNA dynamics that could lead to this differential cargo. Using our data, we explored two possible scenarios. One possibility is that the EV content only reflects the differential expression of miRNAs of E-CD133 cells during normoxia and hypoxia. Of the 39 DE miRNAs in H-EVs vs. N-EVs, only nine reflected the differential expression found in the cells (concurred enrichment) (Figure 3b). A second possibility is that the differential EV content could be related to the selective miRNA loading in the EVs or retention in the cell in each condition. This happened for 75% of the DE miRNAs (Figure 3b).

In hypoxia, 10 miRNAs enriched in H-EVs had no change in the cell expression, suggesting a favored loading. miR-100-3p and miR-34a-5p were even downregulated in H-cells but enriched in H-EVs. On the other hand, two miRNAs were retained in H-cells (despite there being no change in the expression profile), leading to a reduction in H-EVs.

In normoxia, none of the miRNAs enriched in N-EVs (vs. H-EVs) reflected an upregulation in N-cells (vs. H-cells) (Figure 3b). Our data suggested that six miRNAs were enriched in N-EVs due to a favored loading. On the other hand, 10 miRNAs were retained within the cells, then reduced in N-EVs (vs. H-EVs).

We further investigated the differential loading of miRNAs by RT-qPCR. We confirmed that miR-486-5p is enriched in N-EVs when compared to H-EVs (Appendix A). Moreover, we verified that the ratio of miR-486-5p in N-EVs/N-cells is higher than in H-EVs vs. cells (Appendix A). For miR-548d-5p and miR-339-5p, we confirmed the enrichment in N-EVs for only one of the two donors evaluated (Appendix A). Nevertheless, we also found that in normoxia, the EVs are more enriched with these miRNAs (vs. cells) than in hypoxia (Appendix A), reinforcing the hypothesis that the favored loading is important for the dynamics of these miRNAs.

Our data suggest that selective loading is an important pathway to determining the miRNA cargo of EVs. In this work, we identified miRNAs enriched in EV that had been also found as preferentially loaded into EVs in previous studies from literature, such as miR-150-5p; miR-148a-3p, miR-181a-2-3p, miR127-3p and miR-493-5p [43], miR-486-5p, miR-432-5p and miR-216a-5p [44]. The selective loading of miRNAs may be mediated by different mechanisms, such as by RNA binding proteins (e.g., SYNCRIP), which can be regulated by protein post-translational modifications), and by interaction with membrane pathway elements, such as nSMase2 and VPS4 [45,46,47]. miR127-3p and miR-493-5p (found here as N-EVs enriched vs. N-cell), for example, were previously characterized as containing motifs for SYNCRIP and were preferentially loaded into murine hepatocyte-derived EVs [43]. Then, it might be interesting to further investigate which pathways might influence the loading of miRNAs in E-CD133 cells in the future, and if these pathways change according to culture condition.

### 2.4. miRNA EV Cargo under Normoxia and Hypoxia Vary According to Cell Type

Considering the importance of the new data generated with the characterization of miRNAs present in E-CD133 cells and their EVs, both in normoxia and hypoxia, we compared our dataset with a previous study using ECFCs and EVs by Dellet et al. (2017) [44]. For comparison, we used our data from miRNA identified in E-CD133 N-cells and N-EVs. Dellet et al. (2017) also isolated adherent mononuclear cells from hUCB but did not perform CD133^+^ enrichment. After 4 weeks in culture, the ECFCs had a cobblestone morphology and were CD31^+^, CD146^+^, CD14^−^ and CD45^−^ [44], compatible with the expected for ECFCs [31]. Although the use of CD133^+^ as a marker of endothelial progenitors is controversial [31], the methodology used here (enrichment of CD133^+^ mononuclear cells from hUCB followed by expansion in endothelial culture medium) led to an immunophenotype similar to ECFCs.

Around 75% of the miRNA identified in E-CD133 N-cells and N-EVs were also found in ECFCs (Appendix A). On the other hand, 31 miRNAs had not been previously found by Dellet et al.: seven were identified only in E-CD133 cells, eight only in EVs, and 15 were found both in cells and EVs (Appendix A). Among them, miRNAs let-7f-5p, mir-12136, mir-181b-5p, mir-3135b, and mir-484 were enriched in EVs in comparison to the cells (Figure 3a). Their presence in E-CD133 N-EVs and N-cells was confirmed by RT-qPCR (Appendix A), and their putative targets will be discussed later. Additionally, we found in E-CD133 N-EVs miRNAs that may be involved in the control of cancer cell growth, migration and/or maintenance of the stem cell-like phenotype, e.g., miR-4516 [48,49], miR-4286 [50,51], miR-150-5p [52], miR-1275 [53]; angiogenesis, e.g., miR-4286 [54]; osteogenesis, e.g., miR-4286 [54], miR-150-5p [55], miR-548d-5p [56]; and wound healing, e.g., miR-150-5p [57]. Moreover, miR150-5p-enriched exosomes derived from bone marrow mesenchymal stromal cells also demonstrated a protective effect in cerebral ischemia/reperfusion injury [58].

Next, we compared our results with other works that evaluated the miRNA content of N- and H-EVs [59,60,61]. It is noteworthy that the analyzed studies used different cells to produce the EVs and had differences in the methodologies and parameters of analysis. The data obtained demonstrate that the differentially enriched miRNAs in EV isolated from different cell types in normoxia and hypoxia are quite diverse (Appendix A). Only 11 miRNAs were identified as differentially enriched in N-EVs and H-EVs in more than one study. Among the 39 differentially enriched miRNAs found in our study, only seven (miR-143-3p, miR-192-5p, miR-148b-3p, miR-339-5p, miR-125a-3p, miR-370-3p, miR-486) were also identified in at least one of the other works. This suggests that these miRNAs found as DE in H-EVs in more than one cell type might be related to a general hypoxic signature, rather than a specific cell type response.

Here, we identified in E-CD133 cells and EVs the differential loading in EVs or retention in the cells of miRNAs found as differentially enriched in the N-EV and H-EVs from H9C2 [59] and BMSCs from mouse [61] and human [60] (Appendix A). miR-143-3p, miR-148b-3p e miR-192-5p were upregulated in EVs from H9C2 cells under hypoxia [59]. We observed that they are retained in E-CD133 cells under normoxia, but not in hypoxia and that this concurs with a differential enrichment in E-CD133 N-EVs or H-EVs. Still, miR-125a-3p, previously found upregulated in H-EVs from BMSCs [61], had favored loading and was enriched in H-EVs from E-CD133 cells (Appendix A).

let-7i-5p and miR-181b-5p, found as enriched in H-EVs from H9C2 [59], had favored loading in EVs from E-CD133 cells under hypoxia, but not under normoxia. miR-10b-3p, miR-193a-5p, miR-199a-3p, miR214-3p, miR30c-2-3p, miR30d-5p and miR-433-3p were also previously identified as enriched in H-EVs from H9C2 [59] and/or from BMSCs [60]. Here, they were upregulated in E-CD133 H-cells but were retained in the cells and did not have favored EV loading. miR-210, known as Hypoxic-miR [9] had the same pattern. Notably, miR-210 was found enriched in H-EVs from H9C2 [59] and mouse BMSCs [61] but was downregulated in H-EVs from human BMSCs [60] (Appendix A).

### 2.5. Differential Enrichment of miRNAs in EVs Highlights a Set of Target mRNAs and Modulation of Specific Signaling Pathways

To further investigate a possible effect of the differential cargo of miRNA in the E-CD133 EVs, we selected the miRNAs enriched in N-EVs and/or H-EVs (vs. cells) with higher CPM (Figure 4a) to perform gene ontology (GO) analysis of their targets. Among the miRNAs analyzed, miR-12136 did not present targets according to the miRNet 2.0 platform, using the parameters described in the materials and methods section. Interestingly, there are few descriptions of miR-12136 in literature. Recently, it was identified during erythropoiesis [62], in adipose-derived stem cells from old or diabetic patients [63], in the secretome of menstrual blood-derived MSC [64]. miR-12136 was also found to be downregulated in EVs from adipose-derived MSC from patients with metabolic syndrome [65] and was enriched in human vascular smooth muscle cells during replicative senescence [66]. In addition, a study demonstrated that patients with obsessive-compulsive disorders (OCD) may have a different methylation pattern in the gene region for hsa-miR-12136. The authors also predicted some targets of miR-12136 (as ZNF891, CREB1, FLRT2, RPS6KA5, MGAT4C, ZNF714, FGF13, FAM221A, SCAI and CLU4B) and showed that they could be related to the regulation of miRNA processing [67]; interestingly, FGF13 was also one of the genes differentially methylated in OCD patients [68]. Our study adds further evidence of the presence of this miRNA in EVs, and reinforces the importance of deepening this area of study in the future.

The top EV-enriched miRNAs had around 2000 targets mRNA for both conditions (N and H). The network of these top enriched miRNAs and their target mRNAs showed a complex structure (Figure 4b,c), with some targets being shared between two to five miRNAs (Figure 4b,c, Appendix A). In N-EVs the only target regulated by four miRNAs (miR-148a-3p, miR-155-5p, miR-484, and miR-486-5p) was the SMAD2, a mediator of the TGF-β signaling pathway. The miRNAs were able to regulate SMAD2 which leads to, for example, modulation of macrophage response [69] and inhibition of proliferation and metastasis of hepatocellular carcinoma [70].

Among the targets regulated by at least four miRNAs found in H-EVs (let-7f-5p, let-7i-5p, miR-181b-5p and miR-486-5p) is the IGF1R, a transmembrane receptor of the tyrosine kinase family, capable of activating different signaling pathways (such as MAPK, PI3K/Akt, JUN) modulating cellular responses such as maintenance of stem cell pluripotency [71]. Among members of the let-7 family of miRNAs, let-7c overexpression has been shown to inhibit IGF1R expression and reduce osteo/odontogenic differentiation of human dental pulp-derived MSCs [72]. Another member of the let-7 family, let-7i has also been shown to regulate IGF1R expression. EVs derived from multiple sclerosis patients contain let-7i, which suppressed the induction of Treg cells through modulation of IGF1R and TGFBR1 [73]. miR-181b was also related to targeting IGF1R in gliomas, consequently inhibiting proliferation, migration, and tumorigenesis [74].

Other targets regulated by let-7f-5p, let-7i-5p, miR-196b-5p, miR-486-5p, and miR-181b-5p are the High mobility group A (HMGA) proteins 1 and 2 (HGMA1, 2), factors capable of modify the structure of chromatin. They are expressed during embryonic development, in stem cells and cancer, and are related to stemness and differentiation [75]. It was also demonstrated that the HMGA2 is higher expressed in CD133^+^ glioblastoma neurosphere cells [76] and that the HMGA2 knockdown in breast cancer cells decreases the number of CD44^+^ and CD133^+^ cells [77]. In our study, we used an endothelial-like progenitor cell population positively selected using the antibody anti-CD133, but during cell culture, the CD133 protein expression is reduced. One possibility is that the miRNAs identified in EVs reflect the cell phenotype change (a reduction in CD133 expression or a reduction in stemness) and that the EVs contribute to this alteration. Additional assays should be performed to confirm whether the miRNAs presented in the EVs are effectively capable of modulating these targets in the cells of interest (directly or indirectly) and to evaluate the cellular responses resulting from this modulation.

To improve our understanding of the pathways modulated by the mRNA targets we performed GO analysis. The results showed that the mRNA targets (from N- and H-EVs miRNAs) are related to cell cycle and senescence pathways, as well as to FoxO, p53, HIF-1, and AMPK signaling pathways. Although many of the terms appear regardless of the condition, there are differences in their respective *p*-values (Figure 5a,b, Appendix A).

It was possible to notice that among the 10 miRNAs enriched in EVs with the highest CPM, six of them are common to hypoxia and normoxia culture conditions (Figure 4a). Analysis of the target mRNAs of these six miRNAs indicates that they are mainly related to senescence and cell cycle (Figure 5c, Appendix A). When we evaluated the targets of miRNAs overrepresented in H-EVs (let-7f-5p, let-7i-5p, miR-181b-5p and miR-196b-5p) we noticed that the main regulated pathway is the HIF-1 signaling pathway (Figure 5d, Appendix A), whereas for the targets of miRNAs most enriched in N-EVs (miR-127-3p, miR-155-5p, miR-148a-3p and, miR-484), we observed the FoxO signaling pathway (Figure 5e, Appendix A). These analyses indicated that although the miRNAs identified in both conditions, regulated mRNAs related to more general cellular functions (as proliferation), those identified in one or other EV highlights the regulation of more specific signaling pathways.

The evaluation of signaling pathways individually related to each miRNA showed, for example, that members of the let7 miRNA family enriched in H-EVs, let-7f-5p and let-7i-5p, regulate mRNA targets of JAK-STAT and p53 signaling pathways, whereas miR-196b-5p targets participate the signaling pathways related to several hormones, MAPK, and HIF-1 (Table 1). On the other hand, when we consider the miRNA enriched in N-EVs, we observed that miR-155-5p and mir-148a-3p regulate targets related to many pathways, among which FoxO, PI3K-Akt, Wnt and TNF signaling pathways (Table 1). The miR-486-5p targets, the only miRNA considered to be differentially expressed between hypoxia and normoxia in E-CD133 EVs samples, have among its 67 targets, six mRNAs (CDK4, FOXO1, SMAD2, SERPINE1, PIK3R1, PTEN) related to cellular senescence (Table 1).

HIF and FOXO signaling pathways are essential pathways related to cell growth and survival. In hypoxia conditions, to prevent cell apoptosis, a hypoxic adaptative response initiates and it depends on the function of the hypoxia-inducible factor (HIF). These factors activate signaling pathways, which stimulate erythropoietin production, and angiogenesis or alter the metabolism, and can be regulated or regulate the expression of several miRNAs [78]. In our data, we did not find HIF among the targets of the most expressed miRNAs, but there are genes related to this pathway, such as IGF1, IGF1R, ALDOA and CDKN1A (Appendix A).

The Forkhead box transcription factors (FoxO 1, 3, 4, 6) are related to a range of cellular processes. When FoxO is phosphorylated by Akt, it accumulates in the cytoplasm, reducing the expression of FoxO-regulated genes, and favoring cell growth, survival, and proliferation. One of the pathways involved in this process is insulin and IGF, which interact with its receptor (IGF1R), activating the PI3K/Akt that regulates FoxO. However, when FoxO is activated in stress conditions (by JNK) or when FoxO is overexpressed there is an increase in apoptosis, negative regulation of angiogenesis, and suppression of proliferation [79,80]. Many microRNAs could also regulate FoxO levels in different cell types [81], for example, miR-155 [82] and miR-486 [83] found in E-CD133 EVs. Future experiments are needed to confirm the activation or inhibition of these pathways in cells treated with EVs and whether this modulation is related to the identified miRNAs.

The data presented in this work, together with those previously described by the group [11], indicate that the EVs derived from E-CD133 have the potential to modulate different cellular behaviors, aiming their use toward regenerative medicine. Different EVs have been used to treat different diseases [84,85,86,87]; however, to date, there is no definition of which EV source is the best and for which disease or injury. Our group has previously shown the potential of E-CD133 EVs in the treatment of cardiac injury, in a rat model of myocardial infarction [88], and in the recovery of renal function, in a rat model of chronic kidney disease [26]. Future experiments will be able to further assess the relationship between miRNAs (and proteins) identified in E-CD133 EVs, their targets and modulated pathways, and the effects already described.

In conclusion, the use of EVs as a possible therapeutic alternative has grown over the years. Many studies have sought to identify the best cell source for collecting EVs, as well as the best culture conditions since any changes can result in modifications in the content and functionality of the EVs. E-CD133 is a promising cell population little explored in the literature so far. We recently demonstrated the potential of E-CD133 and their EVs in the treatment of cardiac and renal lesions [21,26]. Here, we extended the characterization of E-CD133 EVs by analyzing their miRNA cargo. Our data suggest that miRNA content is determined, at least in part, by a selective loading process that changes in response to the culture condition (hypoxia or normoxia). In-depth knowledge of the content of E-CD133 EVs isolated in different microenvironments can help to determine the best conditions to modulate their cargo for specific uses in regenerative medicine.

## 3. Materials and Methods

### 3.1. CD133^+^ Cell Isolation and Expansion

This study was performed following the guidelines for research involving human subjects and with approval from the ethics committee of Pontifícia Universidade Católica do Paraná (approval number 763) and from the ethics committee of Instituto Carlos Chagas, Fiocruz/PR (CAAE number: 48374715.8.0000.5248).

CD133^+^ cells were isolated from three hUCB donors (collected at Maternidade Vitor Ferreira do Amaral, Curitiba, Paraná, Brazil) as previously described by our group [11,23]. Briefly, after collection of hUCB from fresh placentas (with the umbilical cord still attached), the mononuclear cells were isolated by Ficoll-Hypaque density gradient (d = 1.077) (Sigma-Aldrich, Saint Louis, MO, USA). Then, the CD133^+^ cells were isolated using CD133 Microbead human lyophilized kit (Miltenyi Biotec, Bergisch-Gladbach, Germany), according to the manufacturer’s instructions. CD133^+^ cells were plated and cultured using EBM™-2 Endothelial Cell Growth Basal Medium-2 (Lonza, Walkersville, MD, USA) supplemented with EGM™-2 MV Microvascular Endothelial SingleQuots^®^ kit (Lonza, Walkersville, MD, USA), which includes FBS (5%), hydrocortisone, human fibroblast growth factor-Beta (hFGF-B), vascular endothelial growth factor (VEGF), R3-insulin-like growth factor-1 (R3-IGF-1), ascorbic acid, human epidermal growth factor (hEGF), and gentamicin/amphotericin (GA-1000). The cells were maintained in a humidified incubator at 37 °C and 5% CO_2_, with a culture medium change every 3–4 days. When cells reached confluence, they were harvested using trypsin-EDTA (0.05%) and replated at a density of 2–4 × 10^3^ cells/cm^2^. The experiments were performed with cells from passages 5–10, from three donors named E-CD133 cells (E-CD133 1, E-CD133 2 and E-CD133 3). These cells were characterized by immunophenotyping as previously described [11].

### 3.2. Isolation of Extracellular Vesicles

The EVs were isolated from the conditioned medium (CM) obtained from E-CD133 cells cultured under normoxic or hypoxic conditions. Cell cultures with 70–90% confluence were washed twice with PBS and EBM-2 medium supplemented with EGM™-2 MV factors (hydrocortisone, hFGF-B, VEGF, R3-IGF-1, ascorbic acid, hEGF, GA-1000) and FBS (2%) depleted of vesicles (to avoid cross-contamination) was added (10–12 mL per 150 cm^2^ culture flask). The cells were placed in a humidified atmosphere (5% CO_2_, at 37 °C), under either normoxic (21% O_2_) or hypoxic (5% O_2_) conditions. The CM was collected every 24 h, for a total of 72 h. After the last CM collection, the cells from two culture flasks of each sample were harvested and counted.

For each day, the CM was centrifuged to remove cellular debris (700× *g*, 8 °C, 5 min) and apoptotic bodies (4000× *g*, 8 °C, 20 min), and the supernatant was transferred into a new tube and ultracentrifuged (100,000× *g*, 4 °C, 1 h and 20 min). The EVs were recovered from the bottom of the tube with PBS and stored at 4 °C. For each sample, the EVs fractions obtained from the three days of the collection were pooled, diluted in PBS (final volume of 30 mL), and submitted to a second round of ultracentrifugation (100,000× *g*, 4 °C, 2 h) for cleaning. The EV pellet was resuspended in PBS and the protein content was accessed using Qubit^®^ 2.0 (Life Technologies™, Invitrogen, NY, USA) assay. The EV samples were stored at −80 °C until use.

### 3.3. Characterization of Extracellular Vesicles

The E-CD133 EVs were characterized by nanoparticle tracking analysis (NTA) and transmission electron microscopy (TEM). The analysis of size distribution and concentration of EVs was performed using NanoSight LM10 (NanoSight Ltd., Amesbury, UK), equipped with the NTA analytical software (version 3.4, Malvern Panalytical, Malvern, UK), as previously described by our group [11].

For TEM analysis, initially, the EVs were adsorbed in Formvar-coated copper grids for 1 h at room temperature. The grids were then washed with PBS and incubated with a fixing solution (glutaraldehyde 2.5%, sodium cacodylate buffer 0.1 M) for 10 min at room temperature. After washing three times with sodium cacodylate buffer (0.1 M), the grids were stained with 5% uranyl acetate for 2 min, washed with 18.2 MΩ.cm water, and dried at room temperature. The samples were analyzed using a Jeol JEM-1400 Plus transmission electron microscope (JEOL Ltd., Tokyo, Japan) operating at an acceleration voltage of 100 kV.

### 3.4. Cell Culture under Hypoxic and Normoxic Conditions

The miRNA content of E-CD133 cells under hypoxic and normoxic conditions was also analyzed. First, the E-CD133 cells were cultured under normoxic conditions until reaching 70–80% of confluence. The cells were washed twice with PBS, the adequate culture medium with 2% of depleted FBS (as described for the isolation of EVs) was added and the cells were kept under hypoxic or normoxic conditions for 72 h, without culture medium exchange. The cells were harvested and counted, and the RNA content was extracted as described hereafter.

### 3.5. RNA Isolation and Large-Scale Sequencing

The RNA from cells and EV samples was extracted using the Direct-zol™ RNA Miniprep kit (Zymo Research, Irvine, CA, USA), according to the manufacturer’s instructions. The RNA quantification and the size profile were assessed using the Agilent RNA 6000 Pico kit (Agilent Technologies^®^, Waldbronn, Germany) and an Agilent 2100 Bioanalyzer (Agilent Technologies).

For each sample, 25–50 ng of RNA was used to prepare the cDNA libraries for sequencing, using the TruSeq Small RNA Kit (Illumina, Inc, San Diego, CA, USA), according to the manufacturer’s instructions with slight modification. The cDNA was purified by electrophoresis, using the 6% Novex TBE PAGE gel (Novex™, Life Technologies Carlsbad, CA, USA), and instead of cutting between the custom markers, the bands were excised from 145 bp until up to 500 bp. The quality and the quantification of the cDNA libraries obtained were verified using the Agilent High Sensitivity DNA kit (Agilent Technologies^®^) with the Agilent 2100 Bioanalyzer (Agilent), and the KAPA Library Quantification for Illumina Sequencing Platforms (KAPA Biosystems, Roche, Pleasanton, CA, USA), with the LightCycler^®^ 96 (Roche, Pleasanton, CA, USA). The RNA-Seq was carried out in a MiSeq platform (Illumina), using a 1 × 50 bases configuration.

### 3.6. Bioinformatic Analyses

The sequences in fastq format were analyzed by CLC Genomics Workbench^©^ v 20.0 (Qiagen, Hilden, Germany), using the Homo sapiens genome (hg19) for the total RNA mapping and the ncRNA-specific mapping, the database used was obtained from the Ensembl database Homo_sapiens.GRCh38.ncRNA. The sequences were initially trimmed to remove any internal adapter that remained. TruSeq Small RNA adapter sequence: TGGAATTCTCGGGTGCCAAGG, and when the adapter was identified in the sequence, it was removed along with the following sequence (3′ trim). The parameters used for the alignments were the following: mismatch cost (2), insertion cost (3), deletion cost (3), length fraction (0.7), and similarity fraction (0.8). Only uniquely mapped reads were considered in the analysis.

For the miRNA analysis, the database used was the miRbase version 22.1 and the mapping was performed according to the following parameters: Homo sapiens as the prioritized species; Allow length base isomiRs (no), maximum mismatches (2), strand-specific (yes), minimum sequence length (18), maximum sequence length (55), minimum supporting count (1).

### 3.7. Identification of miRNA Targets and Gene Ontology Analysis

To identify the mRNA targets for the miRNAs and their interactions we used the web-based platform miRNet 2.0 [89]. The target genes were selected based only on miRTarBase v8.0, from Homo sapiens information, based on mirBaseID with no specific tissue selection. The GO analysis was performed in g:Profiler [90] using the g:SCS threshold method, a user threshold of 0.05, and not allowing electronic GO annotations. The identification of the signaling pathways possibly modulated by the miRNA targets was based on KEGG (also performed in g:Profiler) where those pathways related to human diseases were excluded from the analysis. For better visualization of results, graphs were made using GraphPad Prism (version 7.05, GraphPad, San Diego, CA, USA).

### 3.8. Quantitative RT-PCR (RT-qPCR)

For RT-qPCR, 56 ng and 1000 ng of total RNA were used for EVs and cell sample preparation, respectively. C. elegans miR-39-3p mimic (miRNeasy Serum/Plasma Spike-In Control, Qiagen) was added to each sample (3.2 × 10^8^ copies/μg of RNA) as a control. Polyadenylation reaction and 1st-Strand cDNA synthesis were performed using a High-Specificity miRNA 1st-Strand cDNA Synthesis Kit (Agilent Technologies), following the manufacturer’s instructions. For qPCR was used the GoTaq qPCR Master Mix (Promega, Madison, WI, USA), the Universal Reverse Primer of the High-Specificity miRNA 1st-Strand cDNA Synthesis Kit (Agilent Technologies), and the appropriate forward primer (Table 2). The qPCR reaction was performed using the QuantStudio™ 5 Real-Time PCR system (Thermo Fisher, Waltham, MA, USA) and the annealing/extension temperature of 60 °C (30 s) for all primers. The data were analyzed with QuantStudio™ Design and Analysis Software version 2.6.0. The samples were evaluated in technical triplicate and the results were plotted as the expression ratio to miR-39-3p mimic expression (Fold to miR-39-3p).

## Figures and Tables

**Figure 1 ijms-23-10066-f001:**
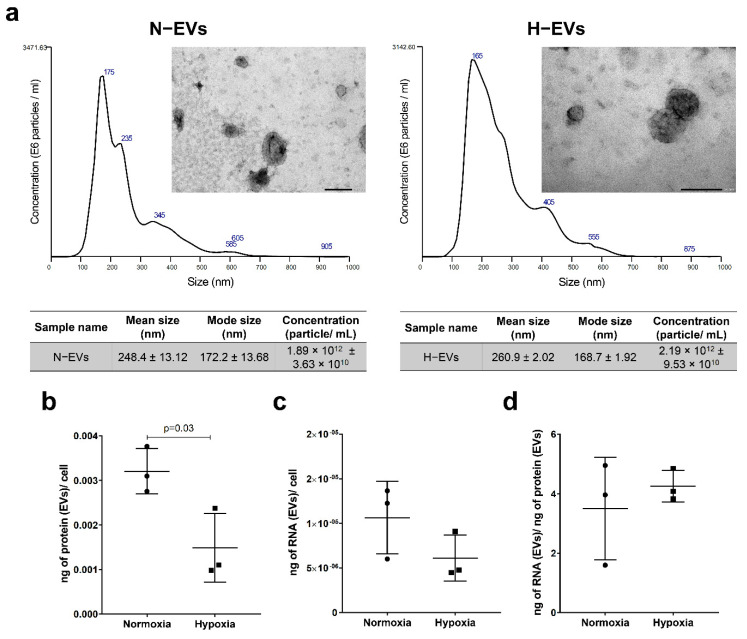
E-CD133 cell and EVs under normoxia and hypoxia. (**a**) Transmission electron microscopy and NTA showing the size distribution of E-CD133 N-EVs and H-EVs (*n* = 3 technical replicates). Scale bar = 200 nm. Analysis of (**b**) the ratio of the protein content of the E-CD133 EVs/cell, (**c**) the ratio of the RNA content of the E-CD133 EVs/cell, and (**d**) the ratio of the RNA content/protein content of the E-CD133 EVs obtained from cells cultivated under normoxia and hypoxia. (Unpaired student *t*-test).

**Figure 2 ijms-23-10066-f002:**
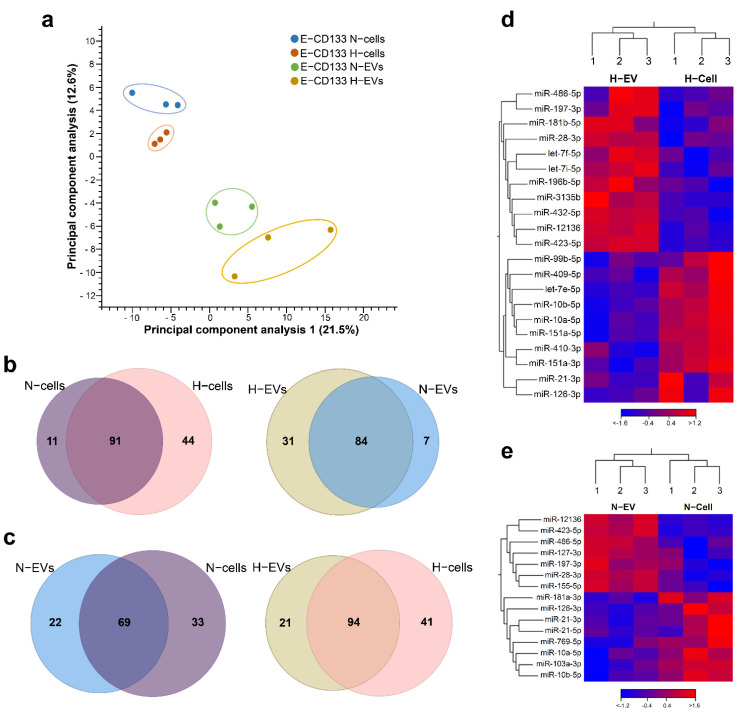
Comparison of miRNAs identified in normoxia and hypoxia E-CD133 cells and EVs. (**a**) Principal component analysis of miRNAs identified in the samples. (**b**) Venn diagrams comparing the identified miRNAs in E-CD133 cells (**left**) or E-CD133 EVs (**right**) in normoxia versus hypoxia culture conditions. (**c**) Venn diagrams comparing the identified miRNAs in E-CD133 cells versus E-CD133 EVs in normoxia (**left**) or hypoxia (**right**) culture conditions. (**d**) Heatmap of miRNAs differentially expressed (FDR ≤ 10%, log2(FC) < −1 and >1) among E-CD133 cells and E-CD133 EV cultured in hypoxia. (**e**) Heatmap of miRNAs differentially expressed (FDR ≤ 10%, log2(FC) < −1 and >1) among E-CD133 cells and E-CD133 EV cultured in normoxia.

**Figure 3 ijms-23-10066-f003:**
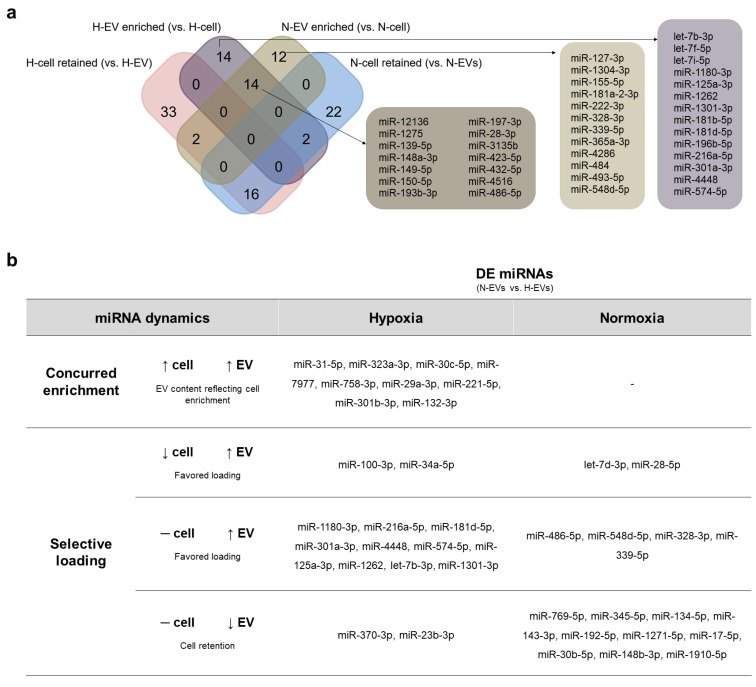
Analysis of miRNAs enriched in E-CD133 EVs or retained in E-CD133. (**a**) Venn chart analysis of the miRNAs found as enriched in the EVs or retained in the cells during hypoxia and normoxia. (**b**) Analysis of the miRNA dynamics that may lead to a DE of miRNAs in H-EVs vs. N-EVs (up arrow = augmentation in miRNA expression/enrichment; down arrow = reduction in miRNA expression/enrichment; dash = no change in miRNA expression/enrichment).

**Figure 4 ijms-23-10066-f004:**
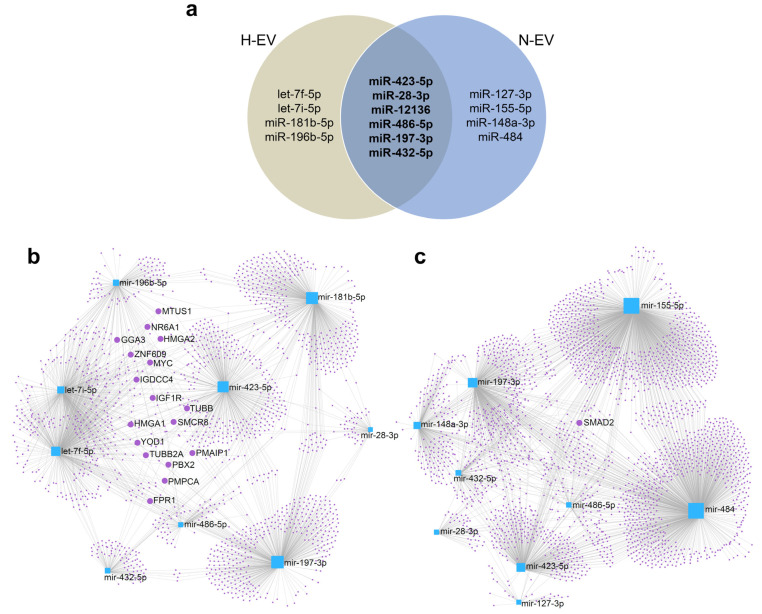
Identification of higher expressed miRNA targets. (**a**) Venn diagram comparing the miRNAs enriched in N-EVs and/or H-EVs (vs. cells) with higher CPM. The miRNAs presented in the two EVs are highlighted in bold. (**b**) Network of top 10 miRNAs enriched in H-EVs (vs. H-cells) and its target mRNAs. (**c**) Network of top 10 miRNAs enriched in N-EVs (vs. N-cells) and its target mRNAs. The highlighted genes are targets of 4 or more miRNAs.

**Figure 5 ijms-23-10066-f005:**
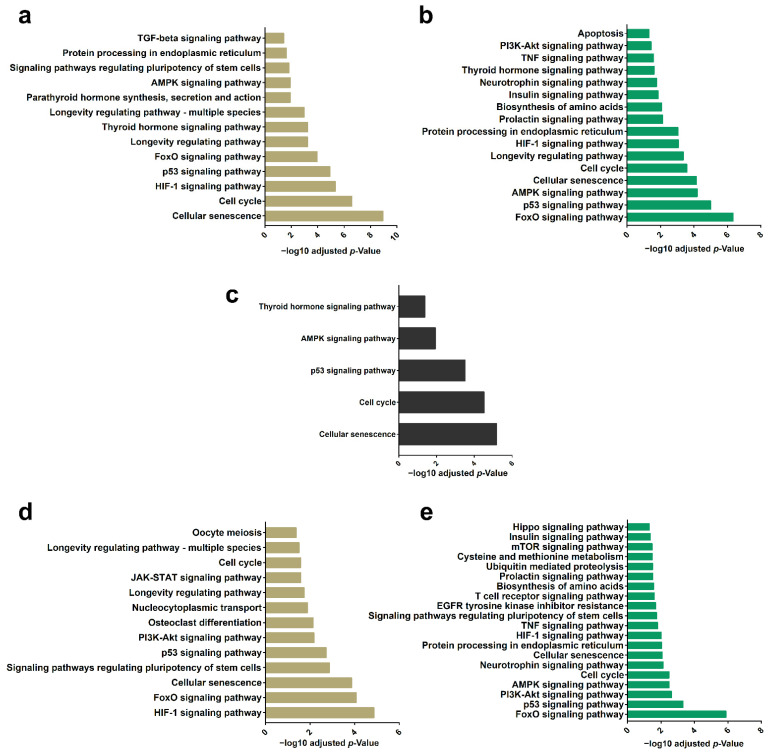
Analysis of miRNA targets and their regulated pathways. (**a**) KEGG analysis of mRNA targets of top 10 miRNAs enriched in H-EVs (vs. H-cells). (**b**) KEGG analysis of mRNA targets of top 10 miRNAs enriched in N-EVs (vs. N-cells). (**c**) KEGG analysis of mRNA targets of the six miRNAs more expressed in H- and N-EVs in comparison with cells. (**d**) KEGG analysis of mRNA targets of the four miRNAs more expressed in E-CD133 EVs in comparison with E-CD133 only in hypoxia condition. (**e**) KEGG analysis of mRNA targets of the four miRNAs more expressed in E-CD133 EVs in comparison with E-CD133 only in normoxia condition.

**Table 1 ijms-23-10066-t001:** KEGG pathways identified from targets for each miRNA enriched in E-CD133 EVs.

Enrichment	miRNA	nº Targets	KEGG Pathway	Adjusted *p*-Value
E-CD133 EV enriched (hypoxia only)	let-7f-5p	397	KEGG:04115—p53 signaling pathway	0.003691
			KEGG:04630—JAK-STAT signaling pathway	0.007606
	let-7i-5p	334	KEGG:04550—Signaling pathways regulating pluripotency of stem cells	0.00683
			KEGG:04630—JAK-STAT signaling pathway	0.020841
			KEGG:04115—p53 signaling pathway	0.038995
	miR-181b-5p	375	KEGG:03013—Nucleocytoplasmic transport	0.015346
			KEGG:04140—Autophagy—animal	0.033046
	miR-196b-5p	150	KEGG:04725—Cholinergic synapse	0.000757
			KEGG:04261—Adrenergic signaling in cardiomyocytes	0.000777
			KEGG:04915—Estrogen signaling pathway	0.0031
			KEGG:04921—Oxytocin signaling pathway	0.007137
			KEGG:04912—GnRH signaling pathway	0.018242
			KEGG:04210—Apoptosis	0.020724
			KEGG:04371—Apelin signaling pathway	0.023727
			KEGG:04010—MAPK signaling pathway	0.030375
			KEGG:04720—Long-term potentiation	0.032404
			KEGG:04922—Glucagon signaling pathway	0.039067
			KEGG:04066—HIF-1 signaling pathway	0.043144
			KEGG:04218—Cellular senescence	0.049952
E-CD133 EV enriched (normoxia only)	miR-155-5p	904	KEGG:04917—Prolactin signaling pathway	3.01 × 10^−5^
			KEGG:04668—TNF signaling pathway	3.07 × 10^−5^
			KEGG:04068—FoxO signaling pathway	0.000133
			KEGG:04660—T cell receptor signaling pathway	0.00054
			KEGG:04550—Signaling pathways regulating pluripotency of stem cells	0.000632
			KEGG:04218—Cellular senescence	0.000873
			KEGG:04657—IL-17 signaling pathway	0.006472
			KEGG:04620—Toll-like receptor signaling pathway	0.006688
			KEGG:04380—Osteoclast differentiation	0.009192
			KEGG:04066—HIF-1 signaling pathway	0.015699
			KEGG:04152—AMPK signaling pathway	0.016785
			KEGG:04151—PI3K-Akt signaling pathway	0.026386
			KEGG:04659—Th17 cell differentiation	0.032336
			KEGG:04120—Ubiquitin-mediated proteolysis	0.045985
			KEGG:04510—Focal adhesion	0.049545
	miR-148a-3p	213	KEGG:04390—Hippo signaling pathway	0.000604
			KEGG:04550—Signaling pathways regulating pluripotency of stem cells	0.001656
			KEGG:04068—FoxO signaling pathway	0.005113
			KEGG:04151—PI3K-Akt signaling pathway	0.005314
			KEGG:04218—Cellular senescence	0.019607
			KEGG:04110—Cell cycle	0.023304
			KEGG:04612—Antigen processing and presentation	0.02659
			KEGG:04310—Wnt signaling pathway	0.032609
			KEGG:04115—p53 signaling pathway	0.036182
	miR-484	891	KEGG:01230—Biosynthesis of amino acids	0.011065
			KEGG:03010—Ribosome	0.033447
E-CD133 EV enriched (both hypoxia and normoxia)	miR-486-5p	67	KEGG:04218—Cellular senescence	0.001365
	miR-423-5p	343	KEGG:04110—Cell cycle	0.010135
			KEGG:01230—Biosynthesis of amino acids	0.042544
	miR-432-5p	94	KEGG:04911—Insulin secretion	0.047795

**Table 2 ijms-23-10066-t002:** Forward primer sequences for miRNA RT-qPCR expression analysis.

Name	Forward Primer Sequence
hsa-miR-100-5p	AACCCGTAGATCCGAACTTGTG
hsa-miR-10b-5p	TACCCTGTAGAACCGAATTTGTG
hsa-miR-99b-5p	CACCCGTAGAACCGACCTTGCG
hsa-let-7f-5p	TGAGGTAGTAGATTGTATAGTT
hsa-miR-12136	GAAAAAGTCATGGAGGCC
hsa-miR-127-3p	TCGGATCCGTCTGAGCTTGGCT
hsa-miR-486-5p	TCCTGTACTGAGCTGCCCCGAG
hsa-miR-3135b	GGCTGGAGCGAGTGCAGTGGTG
hsa-miR-484	TCAGGCTCAGTCCCCTCCCGAT
hsa-miR-423-5p	TGAGGGGCAGAGAGCGAGACTTT
hsa-miR-181b-5p	AACATTCATTGCTGTCGGTGGGT
hsa-miR-339-5p	TCCCTGTCCTCCAGGAGCTCACG
hsa-miR-548d-5p	AAAAGTAATTGTGGTTTTTGCC
cel-miR-39-3p	TCACCGGGTGTAAATCAGCTTG

## Data Availability

Not applicable.

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
