# Peer review of "Selective Loading and Variations in the miRNA Profile of Extracellular Vesicles from Endothelial-like Cells Cultivated under Normoxia and Hypoxia"

_ijms, 2022, doi:10.3390/ijms231710066_

Round 1

Reviewer 1 Report

The authors present a work based on RNA-seq of miRNA in CD133+ endothelial-like cells derived from umbilical cord blood and their respective extracellular vesicles under 2 conditions, normoxia and hypoxia.

Overall, the manuscript is well written, but the work has a major caveat: There are no “follow-up” experiments that can confirm i) the increased or decreased expression of important miRNAs identified in the study, ii) the biological effect on cells of enriched EVs on targets and/or pathways identified in this study. The lack of confirmation of the increased or decreased expression of some miRNAs identified by sequencing is puzzling since the authors claim to confirm the identity of the miRNAs by RT-PCR, which is the gold standard to confirm mRNA/miRNA expression changes.

 The introduction addresses the state of the art and the authors use 18 references. However, only 2 of the references were published in the last 3 years. The authors could update the introduction and also add some information to help readers understand the importance of ECFCs and the importance of CD133+ phenotype when compared with the more frequently used CD31+, CD146+, CD14-, CD45- phenotype.

The paragraph on lines 79 to 88 is a bit confusing. The first time I read it I assumed that the authors were compiling information on ECFCs. It was when I read the manuscript for a 2nd time that I realized that the authors were also presenting their own results. I would rearrange this paragraph and place the results, present in the 2nd sentence, first and then discuss them.

 Figure 2a does not support the authors claim that PCA segregates samples according to condition. The 2 first components only account for 34.1% of the overall variation, which is enough to observe a clear segregation between samples from cells and from EVs but is insufficient to segregate samples from normoxia and hypoxia. I would ask the authors to perform a PCA with data from EV samples and another PCA with data from Cell samples and then observe if there is segregation of samples according to conditions and add the graphs as supplemental information.

Throughout the manuscript, authors state that use RT-PCR to identify the presence of the miRNA (lines 122, 170-171,etc.). Why couldn’t authors confirm the increase/decrease in expression of the miRNAs? This is a missed opportunity to confirm the sequencing results.

 The sentence in lines 147 and 148 must be revised since the reduction of PTEN expression leads to the activation of the AKT pathway.

 The sentence in lines 200-2002 is confusing. The authors mention that “We have identified…” . This means “in previous works of the group” or “the authors found in the literature”? Also the authors reference literature from 2016 and 2017. Why not use more recent works such as this one: https://www.cell.com/molecular-therapy-family/nucleic-acids/fulltext/S2162-2531(22)00067-1 ?

Clarify the statements “…miR-12136 did not show any targets.” And “The authors also indicated some targets of miR-12136” in the paragraph in lines 268-282.

In line 311 there is a typo, HGM2 instead of HGMA2.

The authors in lines 313 to 316 seem to conclude from the regulation network presented in figure 4b that HGMA2 in under expressed. However, there are direct and indirect regulation effects considered in the major databases. So, a particular miRNA can indirectly result in an increase of expression of a particular gene. That’s why it is important to conduct “follow-up” experiments in order to confirm expression changes in HGMA2, SMAD-2 or IGF1R, for example, in cells exposed to EVs.

Author Response

Point 1: Overall, the manuscript is well written, but the work has a major caveat: There are no “follow-up” experiments that can confirm i) the increased or decreased expression of important miRNAs identified in the study, ii) the biological effect on cells of enriched EVs on targets and/or pathways identified in this study. The lack of confirmation of the increased or decreased expression of some miRNAs identified by sequencing is puzzling since the authors claim to confirm the identity of the miRNAs by RT-PCR, which is the gold standard to confirm mRNA/miRNA expression changes.

Response: Thank you very much for your consideration and concerns. We have carefully reviewed our manuscript and addressed the minor and major comments. Your suggestions were very valuable and certainly contributed to enriching the data and the discussion of our work.

Our objective in this article was to characterize the content of miRNAs in EVs derived from E-CD133 cells under two different culture conditions, hypoxia and normoxia, evaluating similarities or differences in the composition of the EVs. We also sought to use an in silico approach to understand miRNA dynamics between cells and EVs. We believe that these analyzes will help in the choice of miRNAs to be studied in the future, either by our groups or others, and then evaluate the activation of their targets and pathways in the target cells of the therapy with EVs. We understand and agree that it would be very interesting to perform functional analysis to understand the effects of the miRNA enrichment in EVs derived from E-CD133 cells under hypoxia and normoxia. Although this was not our main goal in this first work, it may be explored in a future project. Throughout the manuscript, we have included phrases highlighting the need to confirm the targets of miRNAs and the pathways activated by them in cells (Lines 354-357, 412–414).

Regarding qPCR, we confirmed the presence of several miRNAs identified in RNAseq in EVs, initially focusing on the most expressed ones. As suggested by the reviewer, we complemented the analysis of miRNAs by qPCR to confirm the enrichment of miRNAs in either condition. We noticed a variation in the expression of miRNAs depending on the analyzed donor, however, we confirmed the enrichment profile of some of the miRNAs (Supplementary Figure S3 and discussion in section 2.3. Favoring loading in EVs or cell retention concur to differential N- and H-EVs cargo, indicating selective miRNA sorting to EVs, lines 219-227).

Point 2: The introduction addresses the state of the art and the authors use 18 references. However, only 2 of the references were published in the last 3 years. The authors could update the introduction and also add some information to help readers understand the importance of ECFCs and the importance of CD133+ phenotype when compared with the more frequently used CD31+, CD146+, CD14-, CD45- phenotype.

Response: Thank you for your suggestions. We have improved and updated the references in the Introduction section, including more recent studies.

Regarding CD133, we agree with the reviewer that the reader may need more information about the function and phenotype of these cells. Just to give a brief explanation, our work aimed to select endothelial progenitor cells with a more immature phenotype so that there was a more significant ability to proliferate and differentiate in culture. With successive passages in an inductive medium, these cells acquire a more mature phenotype, presenting a set of markers more characteristic of endothelial cells (cells were positive for CD146, CD309, CD31, and vWF; and negative or had low expression of CD45, CD14, CD34, and CD133). The experiments were carried out with this population of cells that we call expanded CD133+ cells (E-CD133). The CD31+, CD146+, CD14-, CD45- phenotype is characteristic of more differentiated cells with a phenotype closer to adult endothelial cells. It is also important to note that our preclinical studies with a mouse model of acute myocardial infarction showed that E-CD133 was more effective in regenerating cardiac tissue than isolated CD133+ cells that did not undergo expansion.

We have partially replaced and added the following explanation to the introduction:

“EPC and hematopoietic stem cells have a common embryonic precursor sharing various markers, such as CD133 and CD34 [28]. The early EPC phenotype represented by the expression of CD133, among others, showed better proliferative activity and greater distribution of primitive progenitors than other studied populations [29]. Early EPCs, also known as myeloid angiogenic cells (MACs), promote angiogenesis in a paracrine fashion, while late EPCs, also called endothelial colony-forming cells (ECFC), participate more directly in the formation of new vasculature [30,31]. An enriched population of early progenitors can be obtained prior to cultivation by magnetic cell sorting based on the positive selection of CD133+ [32]. Then, during successive passages in inductive culture medium, expanded CD133+ cells lose expression of CD133 and start to acquire a phenotype closer to adult endothelial cells expressing markers such as CD31, vWF, and CD146 [22,28].”

Moreover, we performed a careful and more complete review of the literature and updated our references:

  1. van Niel, G. et al. 2022, doi:10.1038/s41580-022-00460-3.
  2. Debbi, L. et al. 2022, doi:10.1016/J.BIOTECHADV.2022.107983.
  3. Leitolis, A. et al. 2019, doi:10.3390/ijms20061279.
  4. Golan-Gerstl, R. et al. 2022, doi:10.1097/MCO.0000000000000834.
  5. Blijdorp, C.J. et al. 2021, doi:10.1681/ASN.2020081142/-/DCSUPPLEMENTAL.
  6. Fernandes, H. et al. 2022, doi:10.1016/j.omtn.2022.03.018.
  7. Akhmerov, A.; Parimon, T. 2022, doi:10.3390/CELLS11142229.
  8. Grange, C.; Bussolati, B. 2022, doi:10.1038/s41581-022-00586-9.
  9. Carnino, J.M.; Lee, H. 2022, doi:10.1016/BS.ACC.2021.07.008.
  10. Rudraprasad, D. et al. 2022, doi:10.1016/J.EXER.2021.108892.
  11. Makhijani, P.; McGaha, T.L. 2022, doi:10.3389/FIMMU.2022.818538.
  12. Han, C.; et al. 2022, doi:10.1016/J.PHARMTHERA.2021.108025.
  13. Psaraki, A. et al. 2022, doi:10.1002/HEP.32129.
  14. Miyasaki, D.M. et al. 2022, doi:10.3390/IJMS23052521.
  15. Rezaie, J. et al. 2022, doi:10.1016/J.LFS.2021.120216.
  16. Dierick, F. et al. 2021, doi:10.3390/CELLS10061338.
  17. Koutna, I. et al. 2011, doi:10.1007/S00277-010-1058-2/FIGURES/5.
  18. Zhang, Q. et al. 2022, doi:10.1155/2022/4460041.
  19. Schreier, S.; Triampo, W. 2020, doi:10.3390/CELLS9040790.
  20. Jiang, H; et al. 2022, doi:10.3389/FIMMU.2022.824188.
  21. Yue, W. et al. 2016, doi:10.1038/srep31333.
  22. Pomatto, M; et al. 2021, doi:10.3390/IJMS22083851.
  23. Rogers, R.G. et al. 2020, doi:10.3389/FPHYS.2020.00479.
  24. Reed, S.L.; Escayg, A. 2021, doi:10.1016/J.NBD.2021.105445.
  25. Szwedowicz, U.; et al. 2022, doi:10.3390/MOLECULES27041303.

Point 3: The paragraph on lines 79 to 88 is a bit confusing. The first time I read it I assumed that the authors were compiling information on ECFCs. It was when I read the manuscript for a 2nd time that I realized that the authors were also presenting their own results. I would rearrange this paragraph and place the results, present in the 2nd sentence, first and then discuss them.

Response: Sorry for the confusion. We rearrange the paragraph following your suggestion and rewrite the paragraph to make it clearer (Lines 91-99).

Point 4: Figure 2a does not support the authors claim that PCA segregates samples according to condition. The 2 first components only account for 34.1% of the overall variation, which is enough to observe a clear segregation between samples from cells and from EVs but is insufficient to segregate samples from normoxia and hypoxia. I would ask the authors to perform a PCA with data from EV samples and another PCA with data from Cell samples and then observe if there is segregation of samples according to conditions and add the graphs as supplemental information.

Response: Thank you for your suggestion. We performed a new PCA with EV or cell samples. The analysis showed the segregation of EV or cell samples based on the culture condition (hypoxia or normoxia). Notably, the separation by condition (hypoxia or normoxia) was more evident in EV samples than in the cell samples (Supplementary figure S2). The results were included in Supplementary Figure S2.

Point 5: Throughout the manuscript, authors state that use RT-PCR to identify the presence of the miRNA (lines 122, 170-171,etc.). Why couldn’t authors confirm the increase/decrease in expression of the miRNAs? This is a missed opportunity to confirm the sequencing results.

Response: Thanks for your suggestion. The reviewer's considerations are very relevant. Initially, we sought to confirm the presence of miRNAs identified in EVs in RNAseq by qPCR, focusing on those miRNAs most identified in EVs (higher CPMs) and also in other previous studies in the literature (e.g. Dellet et al, 2017).

As suggested, to improve our qPCR analysis, we further evaluated the expression of miRNAs identified as differentially enriched in N-EVs vs H-EVs (miR-486-5p, miR-548d-5p, and miR-339-5p). We used samples from two different donors and performed the analysis of each donor in technical triplicate. We confirmed that the miR-486-5p was more expressed in N-EVs in comparison with H-EVs; miR-548d-5p and miR-339-5p were significantly more expressed in N-EVs only in one of the evaluated donors. However, when we performed the ratio of miRNA expression in N-EVs/N-cells, we observed the favored loading of miRNAs for N-EVs in comparison with H-EVs. These results were described in Section 2.3 (2.3. Favoring loading in EVs or cell retention concur to differential N- and H-EVs cargo, indicating selective miRNA sorting to EVs), lines 219-227, and included in Supplementary Figure S3.

Point 6: The sentence in lines 147 and 148 must be revised since the reduction of PTEN expression leads to the activation of the AKT pathway.

Response: Thank you for the comment. We revised and changed the sentence to make it clearer (lines 161-163).

Point 7: The sentence in lines 200-2002 is confusing. The authors mention that “We have identified…” . This means “in previous works of the group” or “the authors found in the literature”? Also the authors reference literature from 2016 and 2017. Why not use more recent works such as this one: https://www.cell.com/molecular-therapy-family/nucleic-acids/fulltext/S2162-2531(22)00067-1 ?

Response: We are sorry for the confusion. In fact, in this sentence, we compare the data obtained in the current work with previous data from the literature, from other groups. Our objective was to show that some miRNAs identified as preferentially loaded in EVs in our study had previously been found with this same profile (higher expression in EVs versus cells) in other works. We've modified the sentence to make this clearer (Lines 229-232). Also, thank you for the reference suggestion. We added it in our introduction section (Line 48).

Point 8: Clarify the statements “…miR-12136 did not show any targets.” And “The authors also indicated some targets of miR-12136” in the paragraph in lines 268-282.

Response: The analysis performed through the miRNet 2.0 platform did not result in any miR-12136 targets. There are few works in the literature on this miRNA. Only recently, in the work by Schiele et al, 2022 (https://doi.org/10.1038/s41398-022-01996-w) some targets were described. We have revised and amended these sentences to make the text clearer. In addition, we include some of the predicted targets described for miR-12136 (section 2.5. Differential enrichment of miRNAs in EVs highlights a set of target mRNAs and modulation of specific signaling pathways, Lines 303-318)

Point 9: In line 311 there is a typo, HGM2 instead of HGMA2.

Response: Sorry for the mistake. We corrected the sentence.

Point 10: The authors in lines 313 to 316 seem to conclude from the regulation network presented in figure 4b that HGMA2 in under expressed. However, there are direct and indirect regulation effects considered in the major databases. So, a particular miRNA can indirectly result in an increase of expression of a particular gene. That’s why it is important to conduct “follow-up” experiments in order to confirm expression changes in HGMA2, SMAD-2 or IGF1R, for example, in cells exposed to EVs.

Response: We appreciate and understand the reviewer’s concerns. This is an initial study, in which we characterized miRNAs from EVs of E-CD133+ cells under different cell culture conditions. In the sentence highlighted by the reviewer, our objective was to bring some studies that showed how the miRNAs identified in our work can change the cellular behavior of some cell types, seeking to relate to the cell or culture condition used in our study. As indicated by the reviewer, to confirm these hypotheses and the interaction of miRNAs with the mentioned targets, whether directly or not, functional assays are necessary. This is not the objective of the current article, however, the data generated in this study would help to choose future miRNAs/targets for functional studies, not limited to our group. We included some sentences (lines 354-357, 412–414) discussing the requirement of functional assays to confirm the interaction between miRNAs-targets-pathways in the cells exposed to EVs.

Reviewer 2 Report

Comments to Authors

v  The authors presented results of their comprehensive analysis of microRNA expression and network in E-CD133 cells and their isolated vesicles at both normoxia and hypoxia, yet  giving their detailed presented results, no recommendations/suggestions for the use of EV transfer therapeutic potential for any specific use in regenerative medicine. This point was absent in the discussion. If they are not able to make any specific recommendation, at least they could discuss additional needed research to enable such objective.

v  In lines 61-62 the authors stated: “Our group had previously shown that expanded CD133+ cells (E-CD133) significantly improved cardiac function in rats after severe myocardial infarction” Reference [9] should follow this sentence.

v  In line 378, the authors stated, “We recently demonstrated the potential of E-CD133 and their EVs in the treatment of cardiac and renal lesions” RFERENCE??

In contrast to the statement in line 271, that  “miR-12136 did not show any targets”, at least ten top targets for miR-12136 have been identified. The top ten predicted targets of microRNA has-miR-12136 are: ZNF891, CREB1, FLRT2, RPS6KA5, MGAT4C, ZNF714, FGF13, FAM221A, SCAI and CLU4B. Most interestingly, the FGF13 gene targeted by hsa-miR-12136 has been identified as one of the top hits in a previous epigenome-wide DNA methylation screen in OCD (Schiele, et al. Epigenome-wide DNA methylation in obsessive-compulsive disorder. Psychiatry. 2022 Jun 1;12(1):221); (Yue W, et al. Genome-wide DNA methylation analysis in obsessive-compulsive disorder patients. Sci Rep. 2016;6:31333),

Author Response

Point 1: The authors presented results of their comprehensive analysis of microRNA expression and network in E-CD133 cells and their isolated vesicles at both normoxia and hypoxia, yet giving their detailed presented results, no recommendations/ suggestions for the use of EV transfer therapeutic potential for any specific use in regenerative medicine. This point was absent in the discussion. If they are not able to make any specific recommendation, at least they could discuss additional needed research to enable such objective.

Response: Thank you very much for your suggestions. We have carefully reviewed our manuscript and addressed the minor and major comments. This review was very valuable and certainly contributed to enriching the data and our work discussion.

 The data presented in this work and those previously described by the group (Anguslki et al, 2017) indicate that the EVs derived from E-CD133 have therapeutic potential. Different EVs have been used to treat different diseases and, so far, there is no definition of which is the best EV and for which disease. Furthermore, different cell types possibly interact/respond in different ways to the same type of EV. Characterizations of EVs content and in vitro assays can indicate the best effect of EVs, for later in vivo assays and follow-up to clinical trials. We've included a discussion about these possibilities and referenced some previous studies of our group regarding potential therapeutic applications of E-CD133 EVs at the end of the Discussion section (Lines 417-426).

Point 2:  In lines 61-62 the authors stated: “Our group had previously shown that expanded CD133+ cells (E-CD133) significantly improved cardiac function in rats after severe myocardial infarction” Reference [9] should follow this sentence.

Point 3: In line 378, the authors stated, “We recently demonstrated the potential of E-CD133 and their EVs in the treatment of cardiac and renal lesions” RFERENCE??

Response points 2 and 3: Thank you for the observation. We included the references in the sentences indicated by the reviewer (line 73 and line 433).

Point 4: In contrast to the statement in line 271, that  “miR-12136 did not show any targets”, at least ten top targets for miR-12136 have been identified. The top ten predicted targets of microRNA has-miR-12136 are: ZNF891, CREB1, FLRT2, RPS6KA5, MGAT4C, ZNF714, FGF13, FAM221A, SCAI and CLU4B. Most interestingly, the FGF13 gene targeted by hsa-miR-12136 has been identified as one of the top hits in a previous epigenome-wide DNA methylation screen in OCD (Schiele, et al. Epigenome-wide DNA methylation in obsessive-compulsive disorder. Psychiatry. 2022 Jun 1;12(1):221); (Yue W, et al. Genome-wide DNA methylation analysis in obsessive-compulsive disorder patients. Sci Rep. 2016;6:31333),

Response: Thank you for your comment and sorry for the confusion. The analysis performed through the miRNet 2.0 platform did not result in any miR-12136 targets. We cited that the work from Schiele et al, 2022, identified some targets for this miRNA. We changed the text to make it clearer, including the ten predicted miR-12136 targets (from Schiele et al, 2022), and highlighting that the FGF13 was also identified as differentially methylated (section 2.5. Differential enrichment of miRNAs in EVs highlights a set of target mRNAs and modulation of specific signaling pathways, Lines 303-318).

Round 2

Reviewer 1 Report

I thank the authors for their replies and the work performed for this revision.

I still have to point to authors that the results don’t support fully the authors claims regarding the “point 4” of my previous comments and ask authors to change the manuscript accordingly.

I thank the authors for performing separate PCA as suggested. However, the results don't fully support the authors claim in lines 128-129 that: “the samples were grouped [by PCA] according to origin (cells or EVs) and also based on culture condition (hypoxia or normoxia).

In suppl fig.2 f) it is possible to observe that PCA groups EV samples according to condition. However, this is not the case for Cell samples (Suppl fig 2 e) and this must be addressed in the final document. The PCA groups samples by type (fig 2) and by condition only in the EV samples (suppl. Fig 2).

In the PCA of the Cell samples (Suppl Fig 2 e), If the authors rotate the ellipsis that highlights the conditions, the authors will get another set of groups, one with 4 samples and another group with 2 samples. If the authors perform the same rotation of the group ellipsis in fig 2a), they will get another set of groups for Cell samples, one group with 5 samples and another group with one sample. The lack of homogenous groups in PCA happens when, either there are outlier samples in groups or if the differences between samples are too small. Observing fig 2a) I think that the differences in Cell samples datasets are too small to be discriminative for condition using PCA. This does not mean that Cell samples are not different. Your results prove that they are different. However, there are a huge amount of similar gene expression that is picked up by PCA in the Cell sample datasets. Furthermore, the PCA analysis gives support to the main objective of your work, that EV samples are different from cell samples and can be discriminated by PCA for condition.

The caption of suppl fig.2 needs to be updated with information for e) and f). 

Author Response

Dear reviewer, thank you for reviewing our manuscript once again and for your valuable observation.

We included your suggestion in our final manuscript (lines 122-127):

"The analysis of the identified miRNAs showed that there are some differences in the miRNAs content of samples from different donors (Supplementary Figure S2a-d). Principal component analysis grouped samples by type (cells or EVs) (Figure 2a) and by condition (hypoxia or normoxia) in EV samples only (Supplementary Figure S2e-f). Although the cell samples in the two conditions showed few differences, these are not enough to clearly discriminate them using PCA (Supplementary Figure S2e-f). "